# Cross-Contamination of Enrofloxacin in Veterinary Medicinal and Nutritional Products in Korea

**DOI:** 10.3390/antibiotics10020128

**Published:** 2021-01-29

**Authors:** JeongWoo Kang, Md. Akil Hossain, Hae-chul Park, Ok me Jeong, Sung-won Park, Moon Her

**Affiliations:** Veterinary Drugs & Biologics Division, Animal and Plant Quarantine Agency (QIA), 177, Hyeoksin 8-ro, Gimcheon-si 39660, Gyeongsangbuk-do, Korea; mdakil_hossain@yahoo.com (M.A.H.); sungpark@korea.kr (H.-c.P.); nabee@korea.kr (O.m.J.); pasawa@korea.kr (S.-w.P.); herm@korea.kr (M.H.)

**Keywords:** antibiotics, drug residue, poultry egg, poultry meat, public health

## Abstract

Poultry meat and eggs are vital sources of protein for human consumption worldwide. The use of several nutritional and medicinal products, including antibiotics, is crucial for efficient and safe poultry production. Accumulation of drug residues in meat and eggs from inappropriate drug use is a major concern to public health. Recently, enrofloxacin was detected (2.4–3.8 ppb) in edible eggs produced in Jeju Island, Korea. Although the farm from which the enrofloxacin-contaminated eggs were collected did not use enrofloxacin-containing products, they reported extensive use of a nutritional product (NPJ). Accordingly, in this study, we investigated whether enrofloxacin contamination had occurred accidentally in various widely used veterinary pharmaceutical products. Enrofloxacin content (4.57–179.08 ppm) in different lots of the NPJ was confirmed by liquid chromatography-tandem mass spectrometry (LC-MS/MS) analysis. Furthermore, 76 veterinary pharmaceutical products that are widely used in poultry farms in Korea and claim to not contain enrofloxacin were collected and analyzed by LC-MS/MS. Among them, a florfenicol product and a sulfatrimethoprime product were found to contain 3.00 and 0.57 ppm enrofloxacin, respectively. These results suggest that appropriate manufacturing standards are not being followed and that strict monitoring of drug manufacturing is necessary in Korea to avoid drug contamination.

## 1. Introduction

Poultry meat and eggs are important foods for fulfilling the dietary needs of the ever-growing human population. Therapeutic and prophylactic use of some veterinary pharmaceutical products, such as antibiotics, enhances the efficiency of healthy poultry production [1]. However, inappropriate and nonjudicious use of these drugs results an accumulation of toxic and harmful residues in the eggs and meat of treated birds, which affects consumer health by triggering allergic reactions, transmitting antibiotic-resistant microbial infections, exerting carcinogenic effects, disrupting the normal intestinal flora, and inducing mutagenesis and teratogenesis [1,2]. Regulatory agencies of many countries operate residue management programs (e.g., the National Residue Program [NRP] of the United States of America, the NRP of Korea (KNRP), and the National Residue Survey of Australia) for managing the risk of drug residues in animal and plant products [3,4,5]. Enrofloxacin is prohibited from being used in poultry in many countries, including Korea, owing to the potential for antibiotic resistance to develop [6]. When enrofloxacin is administered to some food-producing animals, such as poultry, it can be metabolized to ciprofloxacin [7]. Accordingly, the close relationship between fluoroquinolone drugs in veterinary medicine and their use in human medicine may increase the risk of fluoroquinolone resistance being transferred from animals to humans. Indeed, in some studies, the use of enrofloxacin in poultry production has shown to induce fluoroquinolone resistance in *Campylobacter jejuni*, which can then be transferred to humans and contribute to treatment failure of Campylobacterosis in humans via poultry exposure [8,9].

To reduce this risk of residual enrofloxacin, maximum residue limits (MRLs) for enrofloxacin and its metabolite ciprofloxacin have been established in Europe and other countries for muscle, fat, liver, and milk from several animal species. However, MRLs have not yet been established for poultry eggs. Extra-label use of these drugs or unintentional contamination of feed and medicine for laying hens (e.g., cross-contamination during manufacturing processes or transportation) may be the source of drug residues in eggs for human consumption. Moreover, because the appropriate MRL of enrofloxacin in eggs has not been established, regulatory and global trade issues may arise as countries and markets attempt to enforce a “zero-tolerance policy” for this residue [10,11]. Therefore, elimination of these drugs in eggs should be achieved [11]. Unfortunately, several studies have already demonstrated the presence of different antimicrobials (e.g., enrofloxacin) in farm-produced eggs in developing countries [12,13,14,15].

Enrofloxacin was recently detected (2.4–3.8 ppb) in edible eggs produced in Jeju Island, Korea, at a farm where the hens were not administered enrofloxacin. The provincial authority of Jeju Island requested for an emergency investigation of this issue to the Animal and Plant Quarantine Agency (APQA) headquarters of Korea (Jeju Special Self-Governing Province Animal Protection Division-2768 [September 2020]). The farmers from the farm at which the enrofloxacin-contaminated eggs were collected claimed that they did not use enrofloxacin-containing products; however, they reported extensive use of a nutritional product (NPJ; lot number. 812902). 

Accordingly, in this study, we analyzed the enrofloxacin content of this NPJ in order to identify the cause of enrofloxacin contamination of the eggs from this farm. Additionally, we collected and analyzed 76 veterinary medicinal and nutritional products that are commonly used at poultry farms in Korea and that claim to not contain enrofloxacin. The objective of this study was to investigate whether enrofloxacin may be present in these widely used veterinary pharmaceutical products. Our findings are expected to have a great impact on public health.

## 2. Results

The contents of enrofloxacin in different veterinary medicinal and nutritional products were determined by analyzing samples using liquid chromatography-tandem mass spectroscopy (LC-MS/MS). The analytical method was validated prior to use, and the validation results of the LC-MS/MS method for the analysis of enrofloxacin in NPJ are shown in Table 1. Samples of three different lot numbers (812901, 812902, and 812903) of the NPJ were analyzed by LC-MS/MS to determine whether enrofloxacin cross-contamination had occurred; the results are presented in Table 2. All tested samples for the three lots contained substantial amounts of enrofloxacin. Among the three lots, lot number 812902 contained large amounts of enrofloxacin. Representative chromatograms of enrofloxacin for this sample and the standard solutions are shown in Figure 1. 

Next, we evaluated 76 veterinary medicinal and nutritional products that are widely used in the poultry industry in Korea; the results are shown in Table 3. Among the 76 products, two contained enrofloxacin, and both of these products were nonpenicillin antibiotics. Enrofloxacin was not detected in the tested products for the other three categories. The analytical method used to analyze these 76 products was validated prior to use (data not shown). We then investigated the reasons for contamination in these products. The manufacturing facilities of those enrofloxacin-contaminated products were inspected, and the batch histories of the tested lots of contaminated products were checked by the investigation team of APQA. We found that enrofloxacin products were manufactured in the same production facilities (production line) before manufacturing enrofloxacin-contaminated veterinary pharmaceutical products. Moreover, the manufacturing facilities had not been cleaned properly after manufacturing enrofloxacin products. Thus, adherent enrofloxacin products in the manufacturing facilities were mixed with non-enrofloxacin products manufactured after the enrofloxacin products.

## 3. Discussion

The requirement for manufacturing different drugs at separate manufacturing facilities (production lines) will increase both the complexity and overall cost of the manufacturing process. To keep costs low and manufacturing efficient, more than one drug is manufactured using the same production line at different times during campaign production. This can be a source of cross-contamination because residuals from the first drug can be passed to later drugs [16,17]. Moreover, separate production lines for various active pharmaceutical ingredients are often run in parallel to reduce the cost and time of production. Although this is cost effective, it increases the risk of cross-contamination, in which active ingredients from one line can be carried across to the other production lines, e.g., through the air, on workers’ clothing, or via contaminated equipment. This can place both workers and patients at risk. Importantly, if certain sensitizing compounds, such as penicillins and beta-lactam antibiotics, contaminate other drug production lines, allergic reactions can be triggered, even at low levels. The risks range from inconvenient symptoms (e.g., hives, rash, or itchy eyes) to dangerous immune responses, including anaphylactic reactions, which may be fatal [17]. In this situation, risk-based governmental regulatory programs should be designed and implemented to ensure that veterinary pharmaceutical products are produced, distributed, and used in such a way that foods of animal origin are safe for human consumption. 

Current good manufacturing practice guidelines suggest that there should be separation in time followed by appropriate cleaning in accordance with a validated cleaning procedure [16,18]. Unwanted cross-contamination in veterinary medicine can be effectively avoided by proper sequencing, flushing, and cleaning of medicine manufacturing equipment. Proper sequencing and flushing protocols are very effective at preventing cross-contamination of veterinary medicine [10]. Thus, the APQA concluded that veterinary medicine manufacturing premises should be inspected routinely and that manufacturers should be encouraged to develop proper sequencing, flushing, and cleaning protocols for medicine manufacturing equipment. The APQA also designed a routine sampling plan to evaluate the cross-contamination of veterinary medicines and nutritional products and planned to follow-up on manufacturers and others who have a history of noncompliance. After analyzing and collecting more data from upcoming products, the APQA will prepare some regulations for veterinary medicine manufacturers, which will be included in the Korean Veterinary Good Manufacturing Practice (KVGMP) guidelines and will be strictly implemented. The APQA will conduct regular inspections of manufacturing industries to ensure the implementation of KVGMP and of drug premises to ensure adherence to storage conditions as well as good dispensing practices.

In conclusion, consistent application of KNRP guidelines and the establishment and application of KVGMP will help to maintain the good quality of veterinary pharmaceutical products and will ultimately ensure the avoidance of antibiotic and other drug contamination in these medicinal and nutritional products.

## 4. Materials and Methods

Enrofloxacin was obtained from Sigma-Aldrich (St. Louis, MO, USA). High-performance LC-grade acetonitrile and reagent-grade formic acid were purchased from Merck Millipore (Burlington, MA, USA) and Sigma-Aldrich, respectively. A Milli-Q water purification system (Millipore) was utilized to purify water. A YMC C_18_ (3.0 × 100) mm column (3 μm inner porosity) was equipped onto an LC-MS/MS system (LCMS-8045; Shimadzu Corporation, Kyoto, Japan) to determine the amount of enrofloxacin in veterinary medicinal and nutritional products. For quantification of enrofloxacin in widely used veterinary medicinal and nutritional products, we utilized the same methodologies as reported in our previously published article [19]. The mobile phase was a mixture of (A) 0.1% formic acid in distilled water and (B) 0.1% formic acid in acetonitrile, and a gradient flow was maintained with the flow rate of 0.6 mL/min. The mobile phase was initially allowed to flow with a 90:10 ratio of 90% A and 10% B. The ratio of mobile phase solvents was gradually altered to 0:100 (A:B) over 0.1–3 min, and these proportions were maintained until 3.9 min. The proportions were then changed to 5% A and 95% B from 4 to 4.9 min, followed by reversion to the initial ratio (90% A and 10% B) at 5 min; this composition was used until the end of the acquisition. Five microliters of sample was injected at each time. Mass spectrometry with electrospray ionization was used and maintained in positive ion mode. For quantification, the most intense multiple reaction monitoring (MRM) transition were monitored with a second transition for qualitative confirmation. The chromatographic conditions used in the MS/MS detection and quantification of enrofloxacin were as follows: precursor ion, 360.10 (*m/z*); product ion, 342.25 and 316.20 (*m/z*); cone voltage, 22 and 22 V; and collision energy, 18 and 10 eV. The method was optimized and validated prior to analysis of marketed veterinary medicinal and nutritional products. Enrofloxacin was spiked individually in placeboes of all the products during optimizing and validating the method.

To investigate the incidence of contamination at the farm on Jeju Island, Korea, we initially collected samples from three different lot numbers (i.e., 812901, 812902, and 812903) of the NPJ from the manufacturing company because enrofloxacin was identified in eggs after feeding this nutritional product to chickens at the Jeju Island farm. After LC-MS/MS analysis of the product, we collected 76 veterinary pharmaceutical products that are widely used in poultry farms all over the country and are not supposed to contain enrofloxacin. The 76 products comprised 19 nutritional supplements, 12 penicillin antibiotics, 18 nonpenicillin antibiotics, and 27 general medicines (other than antibiotics). Data are expressed as mean ± standard deviations of the mean.

## Figures and Tables

**Figure 1 antibiotics-10-00128-f001:**
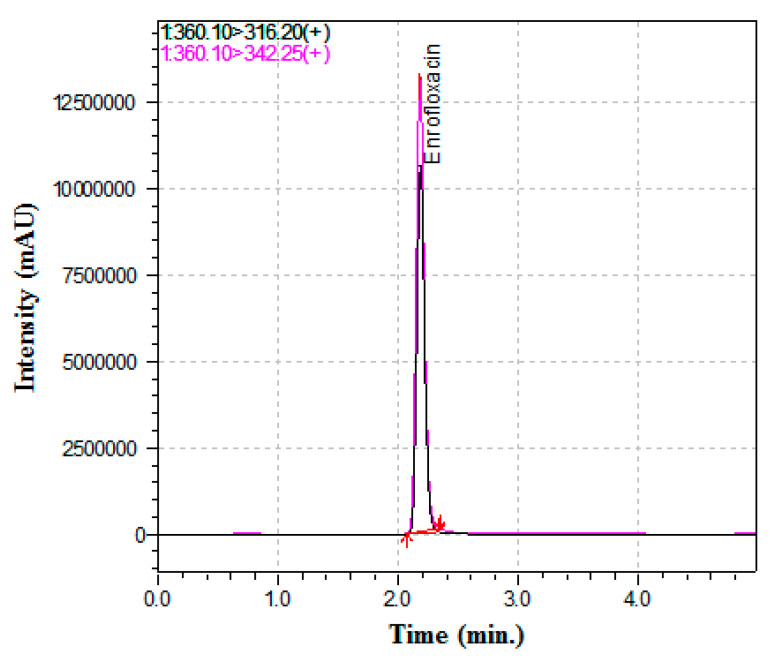
Representative chromatogram of enrofloxacin detected in NPJ. NPJ: veterinary nutritional product that was used in the poultry farm in Jeju Island, Korea, where enrofloxacin was detected in eggs.

**Table 1 antibiotics-10-00128-t001:** Results of the analytical method validation of enrofloxacin in NPJ.

Parameters of Method Validation	Units	Results
Retention time	min	2.32
Linearity (R^2^)	-	0.99
Average recovery	%	99
Coefficient of variation (CV)	%	1.9
Limit of detection (LOD)	ng/g	0.1
Limit of quantitation (LOQ)	ng/g	0.3

NPJ: veterinary nutritional product that was used in the poultry farm in Jeju Island, Korea, where enrofloxacin was detected in eggs.

**Table 2 antibiotics-10-00128-t002:** Quantified amount of enrofloxacin in different lot numbers of the NPJ.

Lot Number	Concentration in ppm (mean ± SD)
812901	4.57 ± 0.04
812902	179.08 ± 0.93
812903	9.71 ± 0.18

NPJ: veterinary nutritional product that was used in the poultry farm in Jeju Island, Korea, where enrofloxacin was detected in eggs; SD: Standard deviation.

**Table 3 antibiotics-10-00128-t003:** Results of enrofloxacin cross-contamination in the widely used veterinary medicinal and nutritional products in Korea except the NPJ those claim to have no enrofloxacin.

Product Category	Number of Tested Product	Number of Contaminated Product	Label Claimed Active Ingredients	Concentration of Enrofloxacin Contaminant (mean ± SD) ppm
Nutritional Supplements	19	0	-	-
Penicillin antibiotics	12	0	-	-
Non-Penicillin antibiotics	18	2	Florfenicol	3.00 ± 0.23
			Sulfatrimethoprim	0.57 ± 0.11
Other medicine (other than antibiotic)	27	0	-	-

NPJ: veterinary nutritional product that was used in the poultry farm in Jeju Island, Korea, where enrofloxacin was detected in eggs; SD: Standard deviation.

## Data Availability

Data will be shared upon request to the corresponding author.

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
