# Peer review of "Cross-Contamination of Enrofloxacin in Veterinary Medicinal and Nutritional Products in Korea"

_antibiotics, 2021, doi:10.3390/antibiotics10020128_

Round 1

Reviewer 1 Report

This paper describes the compact research dedicated to a case study of contamination of the nutritional product with enrofloxacin (ENR). The subsequent monitoring program has been performed in Korea and two other  veterinary medicine products (VMP) were identified with the detectable content of enrofloxacin. This possibility of cross contamination represents the importance message to the producers of VMP and food and veterinary health authorities

Several shortcomings should be addressed by the authors before the publication of this paper: - what criteria were used for the identification of ENR presence by the HPLC-MS/MS methodology? Usually a ratio of different MRM channels are being applied, however, no information is provided by the authors. - Lines 173-174 provide information that precursor ion 277,0 (m/z) and product ions 202.9 and 259.0 were used, however, an example of chromatogram in Figure 1 shows different ratios: 360>316 and 360>342. Please correct.  - Description of Figure 1 should be corrected. There is only one chromatogram there rather than two as follows from the description. - how the authors were dealing with the matrix-effect phenomenon during the analysis of such diverse kinds of matrices?  As I have noticed no internal standard  of ENR has been applied - was ciprofloxacin (CIP) detected as well in edible eggs produced in Jeju Island? Usually CIP is around 10% of ENR and if the applied methodology is sensitive enough you should see some traces of this compound as well.

Author Response

Response:

We like to thank the reviewer for raising this legitimate concern in his comment. We provided more information of the method criteria in this revised version of the manuscript. The added descriptions are highlighted in line no. 17-19 of “Materials and Methods” section in this revised manuscript.

Actually, the precursor ion (360) and product ions (342, and 316) were used for the analysis of enrofloxacin as we have done in our previous study (Figure 2, in J Vet Sci. 2019 Mar;20(2):e15). By mistake incorrect information of precursor ion and product ions were mentioned in the 1st submitted version of the manuscript. In this revised manuscript, we corrected this information in line no. 21 of “Materials and Methods” section.

Description of Figure 1 is corrected in this revised manuscript and highlighted in the figure legend with green color.

Enrofloxacin was spiked individually in placeboes of all the products during optimizing the method. The method was validated prior to the analysis of those products. So, there were no matrix effects in the analysis. The explanation of this issue is mentioned in the line no. 24-25 of “Materials and Methods” section of this revised manuscript.

The Korean National Residue Program (KNRP) uses an analysis method which can simultaneously quantify enrofloxacin and ciprofloxacin. The provincial authority of Jeju Island, a part of KNRP also detected ciprofloxacin in edible eggs produced in Jeju Island, which is believed to be from the metabolism of enrofloxacin in the chicken body. Because, ciprofloxacin is not approved for veterinary use in South Korea, and the veterinary medicine and nutritional product manufacturers are not authorized to purchase ciprofloxacin. Therefore, the objective of our study was to determine the enrofloxacin content (contamination) in medicinal and nutritional products which are used in the poultry farms in South Korea, and are claimed of not having enrofloxacin in it. We only analyzed enrofloxacin content in those medicinal and nutritional products, as enrofloxacin does not metabolize to ciprofloxacin when it is in the form of product.

Reviewer 2 Report

The present manuscript describes an interesting and important topic that must be followed for new studies. Cross-contamination is a problem for many veterinary products and more accurate quality procedures should be operationalized to avoid such situations.

Overall, in my opinion, the manuscript is ready to publication.

Author Response

We appreciate the reviewer for his/her decision.

Reviewer 3 Report

The manuscript targets an indisputably important issue, however, its approach and findings are not adequate to be published as antibiotics research. Nor the analytical chemical method used, or the description of potential hazards of enrofloxacin are new. The number of products tested is too low to be considered as a screening experiment. Interpretation of the results could be enhanced and more detailed. Inconsistent data interpretation occurs sometimes. Just an example: in the Materials and methods section, enrofloxacin was reported to be detected by the precursor ion of 277.00 (m/z); and the product ions of 202.90 and 259.00 (m/z). By contrast, on the Figure 1, the two chromatograms represent the 360.10 -> 316.20 and the 360.10 -> 342.25 mass spectrometric transitions, according to the legend. (Anyway, the legend and caption of this figure also do not correspond to each other.)

Descriptions of authoritive and process control steps do not belong to the scope of Antibiotics. If better described, these issues may be interesting, but not in an antibiotic chemistry journal. A paper dealing with quality control issues may be a better place for the manuscript.

Author Response

Response:

We are thankful to the reviewer for reviewing the manuscript. “Antibiotic resistance and misuse”, and “uses of antibiotics, including on animals and in agriculture” are 2 among 14 scopes of Antibiotics journal as mentioned in (https://www.mdpi.com/journal/antibiotics/about). To our understanding, this manuscript undoubtly belongs to these 2 scopes. Actually, the precursor ion (360) and product ions (342, and 316) were used for the analysis of enrofloxacin as we have done in our previous study (Figure 2, in J Vet Sci. 2019 Mar;20(2):e15). By mistake incorrect information of precursor ion and product ions were mentioned in the 1st submitted version of the manuscript. We corrected this information in line no. 21 of “Materials and Methods” section. Legend of Figure 1 is corrected in this revised manuscript and highlighted with green color.

Round 2

Reviewer 3 Report

The authors improved the manuscript and listed acceptable arguments supporting their choice of journal. The material seems to be acceptable after the revision.